# Combined effects of ambient temperature and food availability on induced innate immune response of a fruit-eating bat (*Carollia perspicillata*)

**Matheus F. Viola[1], L. Gerardo Herrera M.[2]\*, Ariovaldo P. Cruz-Neto[1]**

**1** Laboratório de Fisiologia Animal (LaFA), Departamento de Biodiversidade, Instituto de Biociências, Universidade Estadual Paulista Júlio de Mesquita Filho, Rio Claro, São Paulo, Brazil, **2** Estación de Biología Chamela, Instituto de Biología, Universidad Nacional Autónoma de México, San Patricio, Jalisco, México

\* gherrera@ib.unam.mx

**Data Availability Statement:** All relevant data are within the manuscript and its Supporting Information files.

## Abstract

Resilience of mammals to anthropogenic climate and land-use changes is associated with the maintenance of adequate responses of several fitness-related traits such as those related to immune functions. Isolated and combined effects of decreased food availability and increased ambient temperature can lead to immunosuppression and greater susceptibility to disease. Our study tested the general hypothesis that decreased food availability, increased ambient temperature and the combined effect of both factors would affect selected physiological and behavioral components associated with the innate immune system of fruit-eating bats (*Carollia perspicillata*). Physiological (fever, leukocytosis and neutrophil/lymphocyte ratio) and behavioral (food intake) components of the acute phase response, as well as bacterial killing ability of the plasma were assessed after immune challenge with lipopolysaccharide (LPS: 10 mg/kg) in experimental groups kept at different short-term conditions of food availability (ad libitum diet or 50% food-deprived) and ambient temperature (27 and 33˚C). Our results indicate that magnitude of increase in body temperature was not affected by food availability, ambient temperature or the interaction of both factors, but the time to reach the highest increase took longer in LPS-injected bats that were kept under food restriction. The magnitude of increased neutrophil/lymphocyte ratio was affected by the interaction between food availability and ambient temperature, but food intake, total white blood cell count and bacterial killing ability were not affected by any factor or interaction. Overall, our results suggest that bacterial killing ability and most components of acute phase response examined are not affected by short-term changes in food availability and ambient temperature within the range evaluated in this study, and that the increase of the neutrophil/lymphocyte ratio when bats are exposed to low food availability and high ambient temperature might represent an enhancement of cellular response to deal with infection.

**Funding:** This study was supported by a grant to A. P. C. N and L. G. H. M. from Fundação de Amparo à Pesquisa do Estado de São Paulo (FAPESP Visiting Research Program #2017-17607- 6). A. P. C. N. was supported by a grant from Fundação de Amparo à Pesquisa do Estado de São Paulo (FAPESP, Grant #2014/16320-7), and L. G. H. M. by a grant from the PASPA-DGAPA program of the Universidad Nacional Autónoma de México (#814-2018). M. F. V. was supported by a grant from the Coordenação de Aperfeiçoamento de Pessoal de Nível Superior Tecnológico (CAPES, Grant #88882.434214/2019-01).

**Competing interests:** The authors have declared that no competing interests exist.

## Introduction

Climate and land-use changes might affect animal populations by means of direct abiotic (*e.g.*, changes in temperature and precipitation) and indirect biotic effects (*e.g.*, changes in the abundance of resources, competitors, and predators) [1, 2]. However, the effects of land-use changes are considered to be more immediate than, and often act simultaneously with those caused by climate changes [3]. Thus, the overall magnitude of these extrinsic factors depends primarily on how the effects associated with climate changes, are potentialized by local indirect abiotic effects caused by habitat loss and fragmentation [4]. Sensitivity to these changes can be analyzed by measuring the performance capacity of a given species, that is, its capacity to execute activities that are relevant to its fitness when exposed to these changes. These activities can be associated to aspects related to movement (dynamic performance) or to the ability to regulate physiological processes that mediate the overall responses to these changes, such as thermal performance, regulation of reproduction, growth and the immune system (regulatory performance) [5, 6].

The maintenance of an adequate immune response is crucial to monitor and respond to threats associated with global warming and land-use changes and, thus, are pivotal to maintain physiological homeostasis. Furthermore, how the immune system will cope with the exposure to new pathogens that likely will occur with global changes, might affect the interaction between wildlife hosts and pathogens [7] and, thus, can impact the dissemination of emerging diseases and zoonoses. For example, the stress associated with climate and land-use changes leads to immunosuppression and greater susceptibility to diseases [8]. Studies in populations across different taxa under natural conditions show a correlation between disease susceptibility and global warming in frogs [9] and marine mammals [10]. In turn, land-use change has been associated to increased stress levels and decreased immune performance of reptiles [11].

The effects of climate and land-use change on the immune system can be inferred from results of experimental studies that assessed, respectively, the effects of increasing temperature and decrease food availability on induced innate and acquired immune responses (Tables 1 and 2). Such studies were mostly carried out in mammals and birds, perhaps because of the assumption that activation of immune response is energetic costly and, as such, can have a much broad impact on the energy devoted to other fitness-related activities in endothermic animals. Food-restricted birds show a decrease of cell-mediated response, antibodies levels and acute phase proteins (Table 1), while higher ambient temperatures result in exacerbated fever response, and decreased levels of antibodies and activated macrophages (Table 2). Mammals with limited food availability show attenuated or short fever response, attenuated anorexia, decreased pro-inflammatory cytokines and increased anti-inflammatory cytokines levels, decreased cell-mediated response, and increased neutrophil / lymphocyte ratio (Table 2). However, a decrease in food availability also causes lower increase in total white cell count in some cases, whilst bacterial killing ability and serum antibodies are not affected by limited food availability (Table 1). Furthermore, in mammals, increased ambient temperatures results in attenuated anorexia, lower increase in levels of pro-inflammatory cytokines and haptoglobin, decrease in antibodies levels, and does not affect the magnitude of fever response in most cases (Table 2). Thus, it seems that an increase in temperature and decrease food availability attenuates or exacerbates the magnitude of some components associated with their immune repertoire and, thus, might have a negative impact on the capacity of birds and mammals to maintain an adequate immune response. Although ambient temperature and food availability might simultaneously affect the performance of the immune system, to our knowledge only one study has tested their combined effect on induced immune response [12]. This study shows that commercial broilers raised at 39°C have lower antibodies titer (against sheep

**Table 1.  Overview of studies that evaluated the effect of food availability on induced immune response of birds and non-volant mammals.**

| Species | Immune stimulus | Feeding condition | Time of treatment | Miscellaneous | General results | References |
|---|---|---|---|---|---|---|
| **_Birds_** | | | | | | |
| _Larus cachinnans_ | PHA | Fed ad libitum, fasted and restricted feeding (70%) | 14 days | Adult | Fasting and food restriction decreases CMI. Fasting results in higher decrease in CMI than food restriction. | [13] |
| _Serinus serinus_ | PHA and SRBC | High and low food availability territories | 4 days | Young, Wild nestling | Territories with low food availability results in a lower increase in CMI and lower levels of immunoglobulin. | [14] |
| _Passer domesticus_ | LPS | Fed ad libitum and restricted feeding (30%) | 2 days | Young, closed nestling | Food restriction results in lower increase of acute-phase protein levels. | [15] |
| **_Mammals_** | | | | | | |
| _Ctenomys talarum_ | PHA and SRBC | Fed ad libitum and restricted feeding (~50%) | 12 days | Adult | Food restriction results in lower increase in CMI, and in a higher increase in N/L. BKA, NAB and WBC did not differ among treatments. | [16] |
| _Meriones unguiculatus_ | PHA | Fed ad libitum and fasted | 3 days | Adult, Closed colony | Fasting results in lower increase in CMI and WBC. | [17] |
| _Rattus norvegicus_ | LPS | Fed ad libitum and restricted feeding (50%) | 14 to 28 days | Adult | Food restriction results in absent or attenuated fever response, no anorexia, lower levels of IL-6 pro-inflammatory cytokine and higher levels of IL-10 anti-inflammatory cytokine | [18] |
| _Rattus norvegicus_ | LPS | Fed ad libitum and fasted refeed | 2 days | Adult | Fasting refeed results in reduced LPS-induced Fos expression but do not result in significant attenuation of LPS-induced anorexia | [19] |
| _Rattus norvegicus_ | LPS | Fed ad libitum and fasted | 2 days | Adult | Fasting results in attenuated fever response accompanied by decreased level of some pro-inflammatory cytokine and prostaglandins. | [20] |
| _Mus musculus_ | LPS | Fed ad libitum and restricted feeding (40%) | 28 days | Adult | Food restriction results in lower levels of IL-6, IL-1B and TNF- pro-inflammatory cytokines. | [21] |
| _Mus musculus_ | LPS | Fed ad libitum and restricted feeding (25 and 50%) | 28 days | Adult | Food restriction results in absent or shorter fever response, no anorexia. Increased levels of anti-inflammatory suppressor of cytokine signaling and IL-10 anti-inflammatory cytokine were more pronounced in 50% food restricted group. | [22] |
| _Microcebus murinus_ | LPS | Fed ad libitum and restricted feeding (40%) | 15 weeks | Adult | Food restriction results in a similar magnitude of fever response | [23] |

PHA, phytohemagglutinin; LPS, lipopolysaccharide; SRBC, sheep red blood cells; CMI, cell-mediated response; WBC, total white blood cell count; BKA, bacterial killing ability of plasma; NAB, natural antibodies.

red blood cells; SRBC) than counterpart groups kept at 33°C, independently of food availability. However, those individuals raised at 39°C under intermittent food restriction show higher antibodies titer than counterpart fed ad libitum at the same temperature.

Bats are one of the most diverse groups of mammals and, due to their importance on the maintenance of ecosystem functions, there has been a growing interest in understanding their response to global changes, especially climate and land-use changes [36]. These studies usually favor the use of ecological-based metrics and, as such, they only analyzed the effects of external factors and providing limited insights of the role of intrinsic factors, such as their immune performance. Understanding how immune performance of bats will be affected by such threats is also important because of their epidemiological role. Bats are reservoirs of a number of infectious diseases [37] and understanding how global warming and land-use changes might affect their capacity to deal with zoonotic pathogens is important to understanding the dynamics of dissemination of emerging infectious diseases.

**Table 2. Overview of studies that evaluated the effect of increasing ambient temperature on induced immune response of birds and non-volant mammals.**

| Species | Immune stimulus | Ambient temperature (°C) | Time of treatment | Miscellaneous | General results | References |
|---|---|---|---|---|---|---|
| *Birds* | | | | | | |
| *Anas platyrhynchos* | LPS | 25 and 33 | 14 hours | Adult | Increased ambient temperature results in increased magnitude of fever response. | [24] |
| *Gallus gallus domesticus* | SRBC | 27 and 39 | 7 days | Young | Increased ambient temperature results in lower levels of antibody titer. | [12] |
| *Gallus gallus domesticus* | SRBC | 24 and 35 | 14 days | Young | Increased ambient temperature results in lower levels of antibody titer. | [25] |
| *Gallus gallus domesticus* | SRBC and AEC | 24 and 37 | 21 days | Adult | Increased ambient temperature results in lower levels of immunoglobulin and lower levels of activated macrophages. | [26, 27] |
| *Mammals* | | | | | | |
| *Mus musculus* | LPS | 26 and 33 | 5 days | Adult | Fever response was not affected by ambient temperature. | [28] |
| *Cavia porcellus* | LPS | 20 and 30 | 4 hours | New-born | Fever response was not affected by ambient temperature. | [29] |
| *Rattus norvegicus* | LPS | 22 and 30 | 6 hours | Adult | Febrile response occurs at both ambient temperatures, but the magnitude is higher at milder ambient temperature | [30] |
| *Oryctolagus cuniculus* | LPS | 24 and 32 | 3 hours | Adult | Milder ambient temperature results in biphasic fever, while increased ambient temperature results in monophasic fever | [31] |
| *Sus scrofa domesticus* | LPS | 24 and 30 | 7 days | Adult | Increased ambient temperature results in lower decrease of food intake, lower increase in pro-inflammatory cytokines and haptoglobin levels, and similar magnitude of fever response. | [32, 33] |
| *Cricetulus barabensis* | PHA and KLH | 23 and 32 | 67 days | Female | Increased ambient temperature results in higher increase in CMI and lower levels of immunoglobulin | [34] |
| *Meriones unguiculatus* | PHA | 21 and 30 | 14 days | Female | Increased ambient temperature results in a similar increase in CMI | [35] |

PHA, phytohemagglutinin; LPS, lipopolysaccharide; SRBC, sheep red blood cells; KLH, keyhole limpet hemocyanin; AEC, sephadex stimulation method to recruit abdominal exudate cells; CMI, cell-mediated response.

Studies focused on inflammatory processes of innate immune response are more appropriate to examine the effects of global change on immune response since theses immune defenses clearly impact resistance (and tolerance) of diverse parasites, are most likely to be traded off with other physiological processes, and are evolutionarily conserved [38]. Experimental studies focusing on how bats dealt with immune challenges usually analyzed the parameters associated with the so-called acute phase response. The acute phase response (APR) is the first line of induced defense used by all animals in response to infections, and this response accelerates pathogen elimination and enhances the activation of the adaptive immune system, thus conferring an immediate benefit in controlling infection [39]. The most common technique employed to activate APR is the central or peripherical administration of lipopolysaccharide (LPS) [28, 39, 40], causing the release of proinflammatory cytokines and triggering a suite of behavioral (decrease activity, anorexia) and physiological responses (activation of the HPA axis, fever and leukocytosis) [40, 41]. Moreover, LPS modulates other immunological functions such as bacterial killing ability of plasma, that represents the antimicrobial activity of humoral proteins of the innate immune system [42]. Recent reviews suggested that bats respond selectively to LPS administration, with species-specific differences in the activation of different components of this response [43, 44]. Few studies have analyzed how APR is affected by changes in external factors directly related to global changes. A decrease in food availability delays the metabolic and fever responses of Fish-eating Myotis (*Myotis vivesi*) [45] and fasted Seba´s short-tailed bat (*Carollia perspicillata*) shows increased neutrophil/lymphocyte ratio, lower body mass loss and delayed metabolic response [46].

In this study, we examined how the effect of short-term exposure to high ambient temperature and/or low food availability affects the magnitude of the response of some components of the acute phase response (fever, leukocytosis, the neutrophil/lymphocyte ratio, food intake and body mass), and the bacterial killing ability of fruit-eating bats (*Carollia perspicillata*). By challenging the immune system of these bats under different thermal-feeding treatments, we tested the hypothesis that decreased food availability, increased ambient temperature and the combined effect of both factors would affect bat immune performance. Based on previous results, we predicted that: 1) food restriction would result in a decrease in the magnitude of anorexia and bacterial killing ability, an exacerbation in the magnitude of changes in the neutrophil/lymphocyte ratio and delayed pyrogenic response, but it would not affect white blood cell count; 2) increased ambient temperature would result in a decrease in the magnitude of anorexia, white blood cell count and bacterial killing ability, and an exacerbation in the magnitude of changes in the neutrophil/lymphocyte ratio, but we would not expect changes in the pyrogenic response; 3) the combined effects of food restriction and increased ambient temperature would result in greater attenuation in the magnitude of anorexia, white blood cell count and bacterial killing ability, and a greater exacerbation of changes in the neutrophil/lymphocyte ratio. Considering that mammals generally show attenuated or absent fever when food access is restricted but not when kept at higher ambient temperatures (Tables 1 and 2), we made no predictions about the magnitude of pyrogenic response for the combined effect of food restriction and increased ambient temperature. By linking the combined effects of two external factors that portraits the effects of climate and land-use changes with the response of a regulatory component of performance, we hope to provide insights on the sensitivity of bat's immune system to global changes.

## Material and methods

### Capture and maintenance

Non-reproductive adults of *C. perspicillata* were captured in two municipalities at São Paulo State, Southeastern Brazil: Edmundo Navarro de Andrade State Forest, located in Rio Claro (22˚25'54,2"S 47˚32'11,1"W) and forest remnants of the Federal University of São Carlos—UFSCAR, located in São Carlos (23˚21'32,5"S 46˚15'15,1"W) between September 2019 and July 2021. Bats were captured with mist nets and transported to an external cage (3 x 3 x 3 m), exposed to natural conditions of photoperiod and temperature (21.7 ± 2.8˚C, mean ± s.d. here and thereafter; CEAPLA/IGCE/UNESP) at Universidade Estadual Paulista at Rio Claro. The bats were fed papaya and bananas for five days before experimental trials. The maximum, mean and minimum temperature for Rio Claro in the last 4 years (2017–2020) were 29.3 ± 2.2, 21.6 ± 2.6 and 15.7 ± 3.3˚C respectively (CEAPLA/IGCE/UNESP), while for São Carlos the maximum, mean and minimum temperature were 27.8 ± 2.0, 21.3 ± 2.1 and 16.7 ± 2.4˚C respectively (National Institute of Meteorology). Permits to capture and housing bats were issued by the Instituto Chico Mendes de Conservação da Biodiversidade (ICMBio, process number 66452–1). Ethical permits for this study were issued by the Animal Ethics Committee of the Universidade Estadual Paulista at Rio Claro (Authorization: n˚ 3381).

### Experimental conditions and immune challenge

The immune challenges were conducted in a climatic chamber (3 x 2 x 3 m) with controlled room temperature and photoperiod (12/12 hours). A total amount of 64 bats were transferred to the climatic chamber three days before the immune challenge and kept in individual cages (1 x 1 x 0.5 m). Lights were switched on at 06:00 am and switched off at 06:00 pm. On the first day (72 hours before the immune challenge) sub-cue temperature transmitters were attached

on the back of the bats (see more bellow) and all bats were kept at 27˚C for acclimatization. Forty-eight hours before the immune challenge, thirty-four bats were kept at a room temperature of 27˚C and thirty-four bats were kept at a room temperature of 33˚C. Sixteen bats from each thermal treatment were kept on a restricted feeding regime and sixteen bats on ad libitum feeding regime. The immune challenges were initiated at the beginning of the bat′s activity period (nighttime). We injected 50 µL of a 2.87 mg ml$^{-1}$ solution of LPS (L2630, Sigma-Aldrich, USA) in phosphate buffered saline (PBS; P4417, Sigma-Aldrich, USA) into the scapular area of eight bats assigned to each thermal-feeding treatment. This is equivalent to a dose of 10.01 ± 0.30 mg LPS/kg (mean ± s.d.). In turn, eight bats on each thermal-feeding treatment were injected with 50 ul of phosphate buffered saline solution (control groups). In this way, we conducted 8 experimental trials on the climatic chamber with 8 individuals at a time, and the bats were assigned to only one thermal-feeding treatments. We decided to inject 10 mg LPS/kg in the nighttime because a recent work with *C. perspicillata* showed that higher doses of LPS in the activity period elicit a robust response in some components of the APR, such as reduced food intake and increased N/L ratio [44]. Selected ambient temperatures during the experiments were set close to the lower (27˚C) and upper limits (33˚C) of the thermoneutral zone of *C. perspicillata* (27–35˚C) [47]. These ambient temperatures were above the mean (~22˚C) and maximum (~29˚C) temperatures of the locations where the animals were collected (CEA-PLA/IGCE/UNESP; National Institute of Meteorology). The relative humidity of the climatic chamber at 27 and 33˚C was around 71% ± 2 and 62% ± 3 respectively.

## Food intake and body mass variation

Seventy-two hours before the immune challenge bats were fed banana and papaya ad libitum. Forty-eight hours before and 24 hours after injections, bats were fed with mango nectar (> 95% of the composition in the form of simple sugars—Serigy®) supplemented with 4 mg L$^{-1}$ of hydrolyzed casein (Sigma Aldrich®). We used the approach adopted by Bozinovic et al. [48] to determine the amount of food offered in the ad libitum and restricted treatments. Accordingly, we estimated the food ration (FR) to cover the bat′s daily energy expenses (DEE) according to: *FR (ml day-1) = DEE (kJ day$^{-1}$) / (0.99 ×3.46 kJ ml$^{-1}$)*, where DEE was estimated from the allometric equation derived from field metabolic rate data for bats (FMR kJ day$^{-1}$ = 5.73Mb (g)$^{0.79}$) [49], 0.99 is the digestive efficiency of bats fed nectar [50], and 3.46 kJ ml$^{-1}$ is the energy content of mango nectar of which >95% are in the form of simple sugars. Forty-eight and twenty-four hours before the immune challenge, the amount of food offered in the ad libitum treatment was 2 times the FR, and it was 50% the FR in the restricted treatment. The known amounts of nectar were placed in a feeding device at 06:00 pm and each device was weighed when the food was offered and at 08:00 am to measure food intake 24 hours before and 24 hours after the immune challenge. Twenty-four hours after the immune challenge, all bats of each thermal-feeding treatments received a known amount of ad libitum nectar. We also placed a feeding device in the climactic chamber with a known amount of nectar to measure evaporative loss in each diet-temperature trial. We assessed food intake (FI) changes after the LPS immune challenge in relative terms: ΔFI = (food intake 24 hours after injections—food intake 24 hours before injections) / (food intake 24 hours before injections). Bats of all thermal-feeding treatments were weighed to the nearest 0.1 g (Ohaus Precision Balance, USA) at 18:00 h and 08:00 h 24 hours before injection, immediately before the injection at 18:00 pm, and then 24 hours after injection at 08:00 am and 18:00 pm. We assessed mean body mass (Mb) changes after LPS immune challenge in relative terms: ΔMb = (mean body mass 24 hours after injections–mean body mass 24 hours before injections) / (mean body mass 24 hours before injections) (S1 Fig).

## Bacterial killing ability

Two blood samples of approximately 15 ul were collected with heparinized capillary tubes 24 hours before and 24 hours after the immune challenge. The samples were centrifuged at 3000 rpm for 4 minutes and approximately 7 ul of plasma were collected and stored at -80˚C before analysis. The samples were processed according to the method described by Assis et al. [51]. Plasma samples diluted (1:20) in phosphate-buffered saline solution (3 μL plasma: 60 μL PBS) were mixed with 10 μL of *Escherichia coli* working solution (~$4x10^3$ microorganisms), followed by incubation for 60 min at 37˚C. The positive control consisted of 10 μL of *E. coli* working solution in 63 μL of phosphate-buffered saline solution (no plasma), and the negative control contained only 73 μL of phosphate-buffered saline solution. All samples were incubated under the same conditions. After the incubation period, 500 μL of TSB were added to each sample. The bacterial suspensions were thoroughly mixed and 250 μL of each one was transferred (in duplicate) to a 96 well microplate. The microplate was incubated at 37˚C for 7 hours, at which time the optical density of the samples was measured hourly using a plate spectrophotometer (600 nm wavelength), totaling seven readings. Bacterial killing ability of plasma (BKA) was evaluated at the beginning of the bacterial exponential growth phase (sixth hour of incubation) and was calculated according to the formula: BKA = 1 - (optical density of sample / optical density of positive control). Bacterial killing ability represents the proportion of killed micro-organisms in the samples compared to the positive control. We assessed bacterial killing ability changes after LPS immune challenge in relative terms as: ΔBKA = (BKA 24 hours after injection − BKA 24 hours before injection) / (BKA 24 hours before injection).

## Total white blood cell count and neutrophil / lymphocyte ratio

We collected ∼10 μl of blood from the propatagial vein 24 hours before and 24 hours after injections at around 06:00 pm and prepared two blood smears to estimate total white blood cell count and the neutrophil /lymphocyte ratio changes following Viola et al. [44]. Total and differential white blood cell count in blood smears is the most common method used in bat studies that evaluate the acute phase response after LPS immune challenge [43, 44, 46, 52–55]. We assessed changes in total white blood cell count (WBC) and the neutrophil/lymphocyte ratio (N/L) after LPS immune challenge in relative terms as: ΔWBC = (WBC 24 hours after injection − WBC 24 hours before injection) / (WBC 24 hours before injection); and ΔN/L = (N/L 24 hours after injection − N/L 24 hours before injection) / (N/L 24 hours before injection).

## Body temperature measurement

We measured skin temperature as an approximation of body temperature [56] using Sub-Cue Temperature Transmitters (2.71 ± 0.05 g; Canadian Analytical Technologies, Calgary, Canada) attached in the skin of bats on the scapular region 72 hours before injections. Skin temperature is considered a good estimator of body temperature in bats [45, 57]. Body temperature (Tb) was recorded every hour from 72 hours before and 24 hours after the immune challenge starting at 06:00 pm. We assessed body temperature change 11 hours after the immune challenge (until the lights were turned on at 06:00 am) in absolute terms (ΔTb) by subtracting the hourly body temperature after injections from the respective hourly body temperature 24 hours before injections (S2 Fig). Calibration of the Sub-Cue Temperature Transmitters was performed following Viola et al. [44].

## Data analysis

A factorial ANOVA was used to test the effects of dose, ambient temperature and feeding regime, and their interactions, on the magnitude of changes in food intake, body mass,

bacterial killing ability, total white blood cell count and neutrophil /lymphocyte ratio. A factorial mixed ANOVA with dose, feeding regime and ambient temperature (between subject) and time after injection (within subject) was used to test the effects of these factors, and their interactions, on the magnitude of change in body temperature ($\Delta$Tb). Additionally, a factorial ANOVA was used to test effects of ambient temperature and feeding regime and their interactions on the time at which the LPS-challenged groups reached the maximum increase in $\Delta$Tb. We used Holm-Sidak post-hoc tests for pairwise mean comparisons when the interaction of three factors, two factors and main effects of these ANOVAs were significant in the absence of statistically significant interactions of four factors, three factors and two factors, respectively [58]. Sphericity of factorial mixed ANOVA was assessed with Mauchly's test, and the Greenhouse-Geisser correction was applied when necessary. All variables were checked for normality and homogeneity. In cases where one of the assumptions were violated, data were transformed to meet the assumptions of normality and homogeneity. Accordingly, log transformation was applied on $\Delta$WBC and $\Delta$N/L, and inverse-normal transformation was applied on $\Delta$FI [59]. Studentized residuals test were used to detect significant outliers ($\pm$ 3 s.d.) and removed from the analyses. $\Delta$N/L clearly showed one potential outlier in the group injected with LPS kept at 27˚C on food restriction. All statistical analyses were performed in SPSS 26 for Windows (IBM Corp., Armonk, NY), and a fiducial level of 0.05 was adopted to determine the significance of all comparisons.

## Results

### Changes in food intake and mean body mass

Changes in food intake and body mass as a function of temperature, dose and feeding regime were not affected by the interactions between these factors (S1 and S2 Tables). However, both factors were independently affected by dose ($F_{1,56} = 73.25$, P<0.001, partial $\eta^2 = 0.567$) and feeding regime ($F_{1,56} = 20.79$, P<0.001, partial $\eta^2 = 0.271$), but not by temperature. In fact, changes in food intake were affected independently by dose ($F_{1,56} = 73.25$, P<0.001, partial $\eta^2 = 0.567$) and feeding regime ($F_{1,56} = 20.79$, P<0.001, partial $\eta^2 = 0.271$). That is, regardless of diet or ambient temperature, the magnitude of the decrease in food intake was higher for LPS injected groups compared to those injected with PBS (Fig 1A; S2 Table). Also, regardless of dose or ambient temperature, the magnitude of change in food intake was slightly higher for groups that were on food restriction compared to those that were on ad libitum feeding (Fig 2A; S2 Table). Likewise, there were only significant effects of dose ($F_{1,56} = 56.71$, P<0.001, partial $\eta^2 = 0.503$) and feeding regime ($F_{1,56} = 25.31$, P<0.001, partial $\eta^2 = 0.311$) on mean body mass changes. Regardless of feeding regime or ambient temperature, the magnitude of decrease in mean body mass was higher after LPS injection when compared to the magnitude observed after PBS injection (Fig 1B; S2 Table). Finally, regardless of dose or temperature, the magnitude of the decrease in mean Mb was higher for groups maintained on ad libitum feeding regime (Fig 2B; S2 Table).

### Changes in bacterial killing ability, total white blood cell count and neutrophil/lymphocyte ratio

Neither the interaction effects nor the main effects of dose, ambient temperature and feeding regime on bacterial killing ability and total white blood cell count were significant (Fig 3; S1 Table). In contrast, there was a significant interaction effect of dose×ambient temperature ($F_{1,55} = 8.80$, P = 0.004, partial $\eta^2 = 0.138$) and dose×ambient temperature×diet on neutrophil/lymphocyte ratio ($F_{1,55} = 8.48$, P = 0.005, partial $\eta^2 = 0.134$; S1 Table). In this case, regardless of diet or ambient temperature, all groups injected with LPS showed an increase in neutrophil/

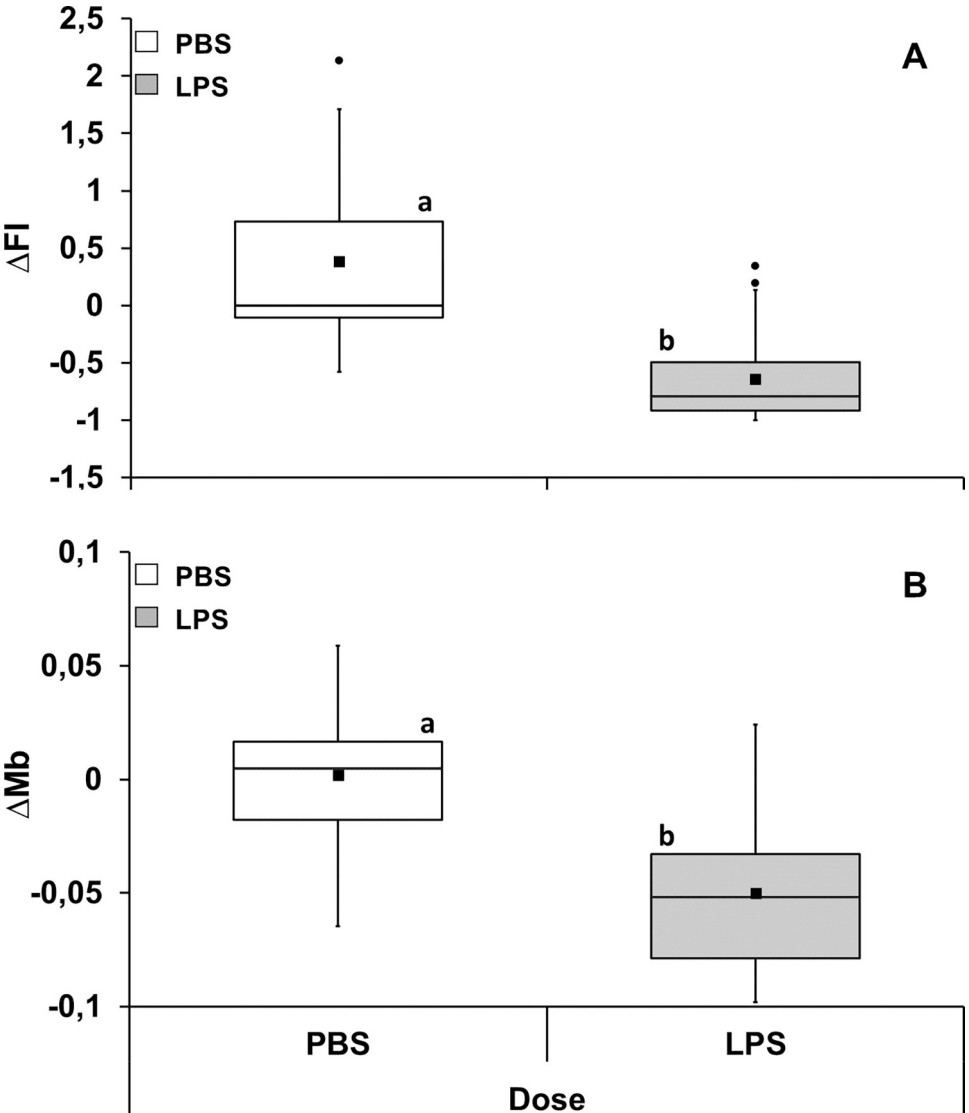

**Fig 1.** Change in food intake (ΔFI—**A**) and mean body mass (ΔMb—**B**) after immune challenge in *Carollia perspicillata*. Data are given for all ambient temperature and feeding regime treatments combined. Different letters: P < 0.001. N = 32 for all groups. Boxplots show medians (horizontal line), means (filled squares), upper and lower limits (vertical lines) and outliers (filled circles).

lymphocyte ratio compared to groups injected with PBS (p≤0.009; Fig 4; S2 Table). The magnitude of increase in LPS-injected bats fed ad libitum at 27˚C was higher than food-restricted bats at the same temperature (p<0.001; Fig 4). Also, the magnitude of increase in LPS-injected bats food-restricted was higher when kept at 33˚C than at 27˚C (p<0.001; Fig 4; S2 Table).

## Changes in body temperature

Hourly body temperature changes (ΔTb) were not affected by the four-factor interaction or any of the three-factor interactions (S3 and S4 Tables). There was only a significant interaction effect of dose×time ($F_{4.61,258.33}$ = 2.80, P = 0.035, partial $\eta^2$ = 0.048) on hourly changes of body temperature, indicating that the magnitude of change over time depends on the dose, but not on the ambient temperature and/or feeding regime. That is, regardless of diet or ambient

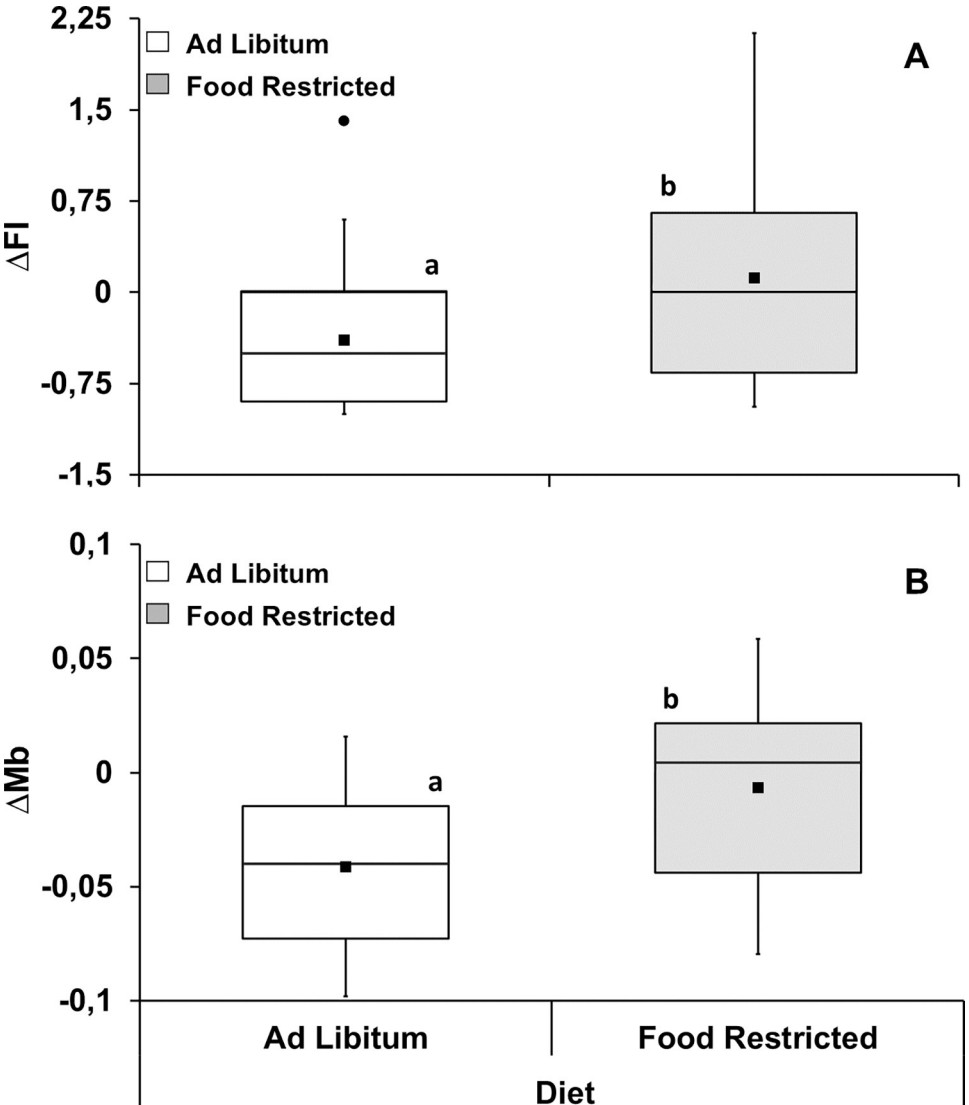

**Fig 2.** Change in food intake (ΔFI—**A**) and mean body mass (ΔMb—**B**) after immune challenge in *Carollia perspicillata* under different feeding regime. Data are given for all injections and ambient temperature treatments combined. Different letters: P < 0.001. N = 32 for all groups. Boxplots show medians (horizontal line), means (filled squares), upper and lower limits (vertical lines) and outliers (filled circles).

temperature, hourly ΔTb of bats injected with PBS did not significantly change with time after injection (P>0.269), but a significant increase in ΔTb of LPS-injected bats occurs four hours after injection compared with the first hour of the respective group (P<0.001; Fig 5; S4 Table). Also, regardless of dose or ambient temperature, no significant difference in ΔTb of bats injected with PBS or LPS was observed between 1 and 2 hours after injections (P>0.05), but ΔTb of LPS-injected bats was higher than those injected PBS between 4 and 6 h after the immune challenge (from 0.27 to 0.9˚C; P<0.02; Fig 5; S4 Table). After the maximum increase in ΔTb of LPS-injected bats, it steadily decreases converging on the ΔTb of PBS-injected bats (P>0.188; Fig 5). The time at which the LPS-challenged groups reached the maximum increase in ΔTb was not affected by ambient temperature or the feeding regime×ambiente temperature interaction. There was a statistical trend of a significant effect of the feeding regime ($F_{5.22,292.28}$

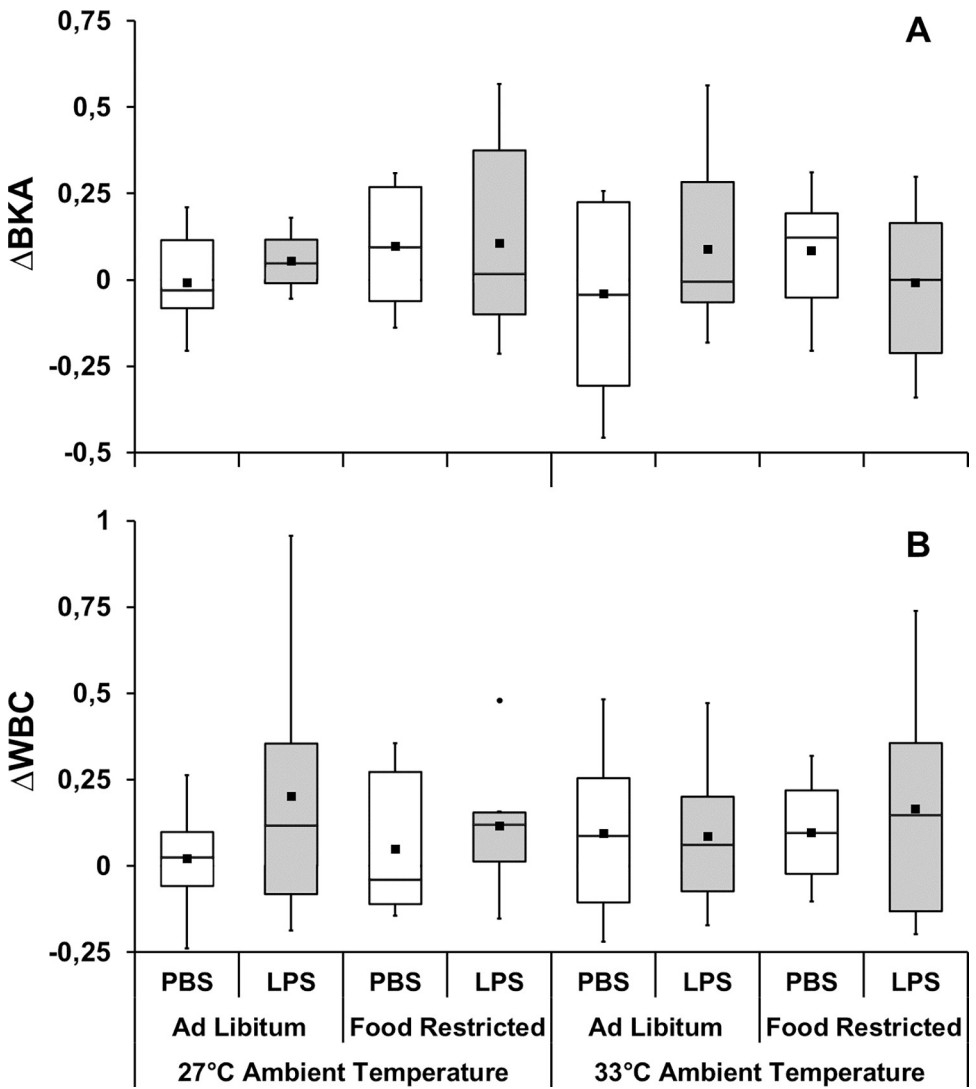

**Fig 3.** Change in bacterial killing ability (ΔBKA—**A**) and total white blood cell count (ΔWBC—**B**) **after** stimulating the acute phase response in *C. perspicillata* kept at different ambient temperatures and feeding regime. N = 5 for the PBS group kept at 33˚C with ad libitum food, 7 for PBS group kept at 27˚C with ad libitum food, and 6 for all other groups (**A**). N = 8 for all groups (**B**). Boxplots show medians (horizontal line), means (filled squares) and upper and lower limits (vertical lines).

= 2.38, P = 0.053, partial η2 = 0.127; S3 Table) indicating that the time to reach the maximum increase in ΔTb was slightly longer for bat kept under food restriction (Fig 6; S5 Table).

## Discussion

To our knowledge this is the first study investigating the combined effects of food availability and ambient temperature on physiological and behavioral components of the acute phase response, as well as on bacterial killing ability of wild vertebrates. We found partly support to our predictions. After the LPS-injection, food intake and total white blood cell count were not affected regardless of food availability or ambient temperature in which the bats were kept. Furthermore, bacterial killing ability was not affected by LPS-injection, food availability, ambient temperature or the interaction of these factors. Increased body temperature was observed

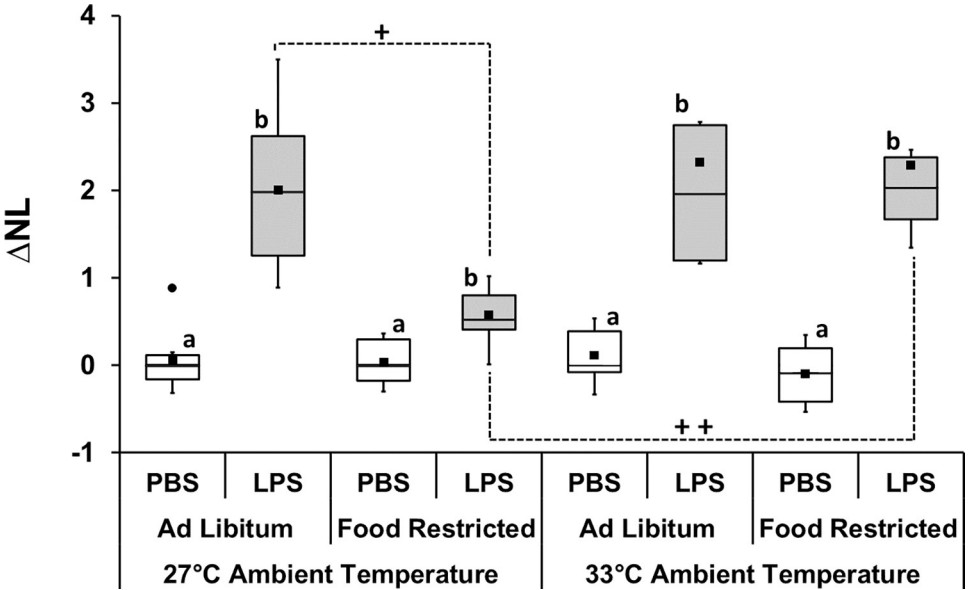

**Fig 4. Change in neutrophil/lymphocyte ratio (ΔN/L) after immune challenge in *C. perspicillata* kept at different ambient temperatures and feeding regime.** Different letters represent significant differences (P ≤ 0.009) between LPS injected groups and PBS injected counterpart. +P ≤ 0.001; ++P ≤ 0.001. N = 8 for all groups except the food-restricted LPS group kept at 27˚ (N = 7). Boxplots show medians (horizontal line), means (filled squares), upper and lower limits (vertical lines) and outliers (filled circles).

regardless of food availability or ambient temperature, but the time to reach the maximum increase in body temperature took longer in bats kept under food restriction. Lastly, the neutrophil /lymphocyte ratio after the LPS-challenge was the only response affected by the interaction between food availability and ambient temperature.

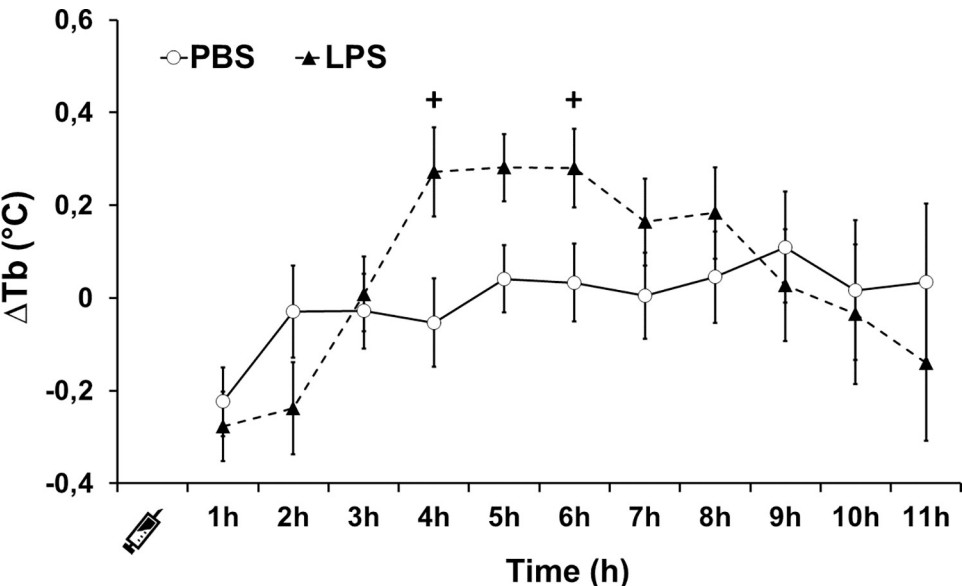

**Fig 5. Changes in body temperature after immune challenge in *C. perspicillata*.** Data (mean ± standard error) are given for all ambient temperature and feeding diets treatments combined. +P < 0.020 for pairwise comparisons between LPS- and PBS-injected bats at the same period after injection. N = 32 for LPS and PBS groups.

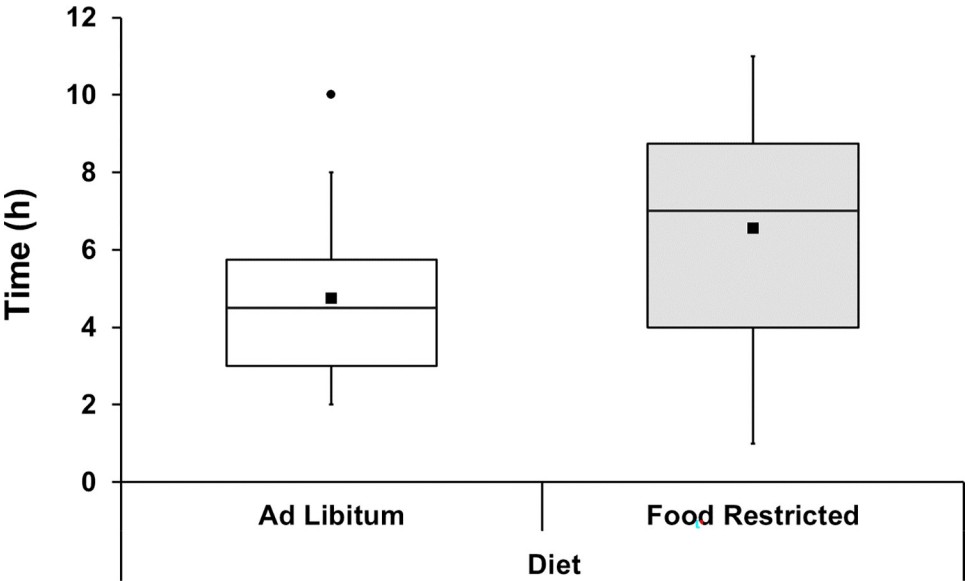

**Fig 6. Time to reach the maximum increase in ΔTb in LPS-injected individuals of *C. perspicillata* fed ad libitum and under food-restriction.** Data are given for all ambient temperature treatments combined. There was a statistical trend (P = 0.053) for a longer time in food-restricted bats. Boxplots show medians (horizontal line), means (filled squares), upper and lower limits (vertical lines) and outliers (filled circles). N = 32 for each group.

The anorexic behavior of LPS-challenged bats was not affected by food availability, ambient temperature or by the combined effect of both factors. In general, mammals increase food intake rather than displaying anorexic behavior after LPS-exposure under conditions of low food availability or higher ambient temperature [18, 22, 33]. The physiological bases behind these findings result from down-regulation of pro-inflammatory cytokines, including those involved in the expression of anorexic disease behavior, resulting in attenuated or absent decrease in food intake [18, 20, 33, 60, 61]. However, it is likely that a longer exposure to these factors and/or to an increased magnitude could alter the down-regulation of pro-inflammatory cytokines and consequently the LPS-induced anorexic behavioral response. For instance, anorexic behavioral response is attenuated after a long-term exposure to higher temperatures (pigs: 7 days at 30°C) and to low food availability (small rodents; 14 to 28 days with a restriction of 25 to 50%) [18, 22, 33]. On the other hand, short-term exposure to low food availability does not result in significant attenuation of anorexia (small rodents: 48 hours of fasting) [19].

Previous studies with small rodents fasted or food-restricted attribute the lower body mass loss after LPS challenge to a lower decrease in food intake and/or attenuated or absent increase in body temperature [18, 20, 22], while a previous study with pigs attribute the lower body mass loss to lower decrease in food intake of animals kept at higher ambient temperature [32, 33]. Regardless of ambient temperature or injection (PBS or LPS), food-restricted bats lost less body mass, reflecting the slight mean increase in food intake of bats under food restriction. Considering that food availability, ambient temperature or their interaction did not significantly affect the reduction of food intake, and that the time to reach the maximum increase in body temperature took longer in bats kept under food restriction, it is not clear to what extent the changes in body mass loss when fed the restricted diet is the result of changes in food intake and body temperature in bats. Individuals of *C. perspicillata* under overnight fasting show similar increase in body temperature and less body mass loss compared to bats fed ad libitum after LPS injection, while individuals of *Myotis vivesi* under overnight food restriction

show delay in body temperatures increase and less body mass loss than bats fed ad libitum [45].

Bacterial killing ability was not affected by LPS injection, diet, ambient temperature or the interaction of these factors. In turn, we found that some individuals increased, decreased or maintained the same bacterial killing ability regardless of the factors to which they were subjected (S2 Table). Our results contrast with those reported in previous studies, which may be explained by a species-specific response and/or by the timing in which the effect was measured. For example, increased bacterial killing ability is reported in frogs and toads between 20 and 24 hours after LPS challenge [62–64], while in turtles it is reported 48 hours after LPS challenge [65]. Increased levels of bacteriostatic acute phase protein (haptoglobin) is present in house sparrow nestlings and mice 20 hours, and 48 hours after LPS challenge, respectively [15, 39].

We found no significant changes in total white blood cell count in response to LPS, ambient temperature or food availability. In turn, we found inter-individual variations, with some individuals increasing, decreasing or maintaining the number of white blood cell regardless of the factors to which bats were subjected (S2 Table). Similar to our findings, one-night of fasting does not affect total white blood cell count nine hours after LPS challenge in *C. perspicillata* [46]. Previous studies with wild small rodents challenged with PHA reported that food restriction does not affect white blood cell response in individuals of *Ctenomys talarum*, while fasted individuals of *Meriones unguiculatus* show a lower increase in total white blood cell count than individuals fed ad libitum (Table 1). To our knowledge, no other studies have investigated the effect of increased ambient temperature on the induced white blood cell response. Further studies should consider investigating the effect of low food availability and increased ambient, as well as their interaction, to better clarify the direction and magnitude of induced white blood cell response across taxa.

Similar to previous studies with bats, LPS triggered an increase in neutrophil/lymphocyte ratio [44, 57, 66, 67], however, the magnitude of this increase was affected by the dose-diet-temperature interaction. Contrary to our predictions, LPS elicited an expected relative increase in neutrophil/lymphocyte ratio only in bats kept at 33˚C under food restriction. LPS activates the HPA axis culminating in a quick glucocorticoids (GCs) release, which in turn can lead to immune cell redistribution, resulting in neutrophilia and lymphopenia. Increase of the neutrophil/lymphocyte ratio is commonly used as stress indicator [68, 69] although in our study the exposure to food-reduction and high temperature led to neutrophilia and lymphopenia only in bats subjected to the immune challenge. During acute phase response, glucocorticoids increase migration of bone marrow-derived neutrophils to the bloodstream, where they are essential in the fight against infections, while redirect traffic of circulating lymphocytes from the blood stream to lymph nodes, spleen, bone marrow and skin, improving the innate and adaptive immune response [70–74]. Thus, it is possible that the combined exposure to a higher ambient temperature and limited food access produced an increase of glucocorticoid levels favoring the circulation of neutrophils to cope with infection.

The LPS-challenge triggered an increase in body temperature regardless of short-term exposure to low food availability and/or high ambient temperature. Body temperature peaked 4 hours after injections, similar to previous studies with bats, small rodents and birds [28, 44, 45, 75–77], however the time to reach its maximum increase took longer in LPS-injected bats that were kept under food restriction. Our findings matched previous work with fish eating bats (*M. vivesi*) kept under food restriction (65%) for one night [45]. The effect of food restriction on body temperature after LPS challenge has been examined in some studies and appears to be related to the duration and magnitude of the restriction, and also to species-specific responses. Lab rats under food restriction (25% restriction for 14 days) show attenuated increase in body temperature but this increase was absent when exposed to a higher and longer

restriction (50% for 28 days) [18]. Grey mouse lemurs (*M. murinus*) under food restriction (40% for 15 weeks) show a similar increase in body temperature after the LPS-challenge [23]. As reported in small rodents, these changes in body temperature might result from the down-regulation of pro-inflammatory pathways and the intensification of anti-inflammatory pathways [22]. Although not strictly comparable due to the time interval of exposure to higher temperatures (Table 2), our findings contrast with previous studies with Pekin ducks (*Anas platyrhynchos*) that show increased body temperature when kept at higher ambient temperature [24], and are similar to those of several studies with mammals showing no effect of ambient temperature on body temperature after an LPS challenge [28, 33, 78, 79]. It seems that in mammals, fevers are regulated to reach the set-point temperature regardless of the increase in ambient temperature, whereas Pekin ducks fail to defend the temperature set-point when kept at higher ambient temperature [24, 80]. Based on the limited work available on bats challenged with LPS, we speculate that short-term exposure to higher ambient temperature does not constrain or exacerbate the increase in body temperature, while short-term exposure to low food availability does not constrain but delayed the time to reach the maximum increase in body temperature. The febrile response prevents pathogen replication and increases the efficiency of immune responses, stimulating both the innate and adaptive arms of the immune system, and it is associated with shortened disease duration and improved survival in most animals [81, 82]. To what extent the delay in the time to reach maximum increase in body temperature affect or not the activation pathways of the bat immune system and consequently the host-pathogen interaction after a short-term exposure to low food availability is not known and requires further investigation.

## Conclusions

Our study offers the first insights into the combined effect of food availability and ambient temperature on LPS-induced acute phase response and bacterial killing ability of bats and how they deal with the infections under these conditions. Overall, our results suggest that low food availability and/or increased ambient temperature do not exacerbate or attenuate most acute phase response components and bacterial killing ability. On the contrary, our results indicate the maintenance of a similar immune response even when this species was subjected to the combined effects of these factors. The fact that the immunological responses of *C. perspicillata* were not affected by short-term changes in food availability and ambient temperature, at least under the range evaluated in the present study, suggests that this species might be able to maintain its regulatory performance to cope with land-use and climate changes. The relative increase in neutrophil trafficking triggered by differential activation of the HPA axis when bats were exposed to high temperature and low food availability indicates that their cellular immune response was enhanced to deal with infection. However, we must be cautious because higher N/L ratios might be detrimental to bat´s health due to their association with excessive levels of reactive oxygen species [83]. Although this result underscores previous observation that this species is resilient to anthropogenic changes [84], we do not rule out that a longer-term exposure to these factors and/or increased magnitude could alter the direction and magnitude of this responses, health status and consequently the host-pathogen interaction. Chronic stress can decrease number and functions of immune cells and /or enhance the effects of immunosuppressive mechanisms leading to immune suppression [85]. Additionally, considering that the evaluated responses may not show their maximum performance in the same time interval, further studies that evaluate the synergistic effect of both factors on the LPS-induced immune response should consider evaluating the effects over a longer period with shorter sampling intervals.

## Supporting information

**S1 Fig. Body mass change (ΔMb) of *C. perspicillata*.** Body mass changes was assessed in relative terms as: body mass change (ΔMb) = (mean body mass after injections–mean body mass before injections) / (mean body mass before injections).
(PDF)

**S2 Fig. Body temperature change (ΔTb) of *C. perspicillata*.** Hourly body temperature change was assessed 11 h after injections in absolute terms by subtracting hourly skin temperature after injections (TAhourly) from the respective hourly skin temperature before injections (TBhourly).
(PDF)

**S1 Table. Result of analyzes for acute phase response components and bacterial killing ability in *Carollia perspicillata*.** Factorial ANOVA analyses was used to examine the effect of dose (PBS and 10 mg/kg LPS), ambient temperature (27˚C and 33˚C), diet (ad libitum feeding and restricted feeding) and the effects of their interactions on food intake change (ΔFI), body mass change (ΔMb), bacterial killing ability change (ΔBKA), total white blood cell change (ΔWBC) and neutrophil/lymphocyte ratio change (ΔNL).
(PDF)

**S2 Table. Changes in acute phase response components and bacterial killing ability of plasma after immune challenge in *Carollia perspicillata*.** FIB: mean food intake before injections; FIA: mean food intake after injections; ΔFI: mean food intake changes after injections; WBCB: mean total white blood cell count before injections; WBCA: mean total white blood cell count after injections; ΔWBC: mean total white blood cell count change after injections; RNLB: mean neutrophil/lymphocyte ratio before injections; RNLA: mean neutrophil/lymphocyte ratio after injections; ΔRNL: mean neutrophil/lymphocyte ratio change after injections. BMB: mean body mass before injections; BMA: mean body mass after injections; ΔMb: mean body mass change after injections. N = 7 for ΔRNL of food restricted group at 27˚C injected with LPS. N = 8 for all other groups.
(PDF)

**S3 Table. Results of body temperature analyzes in *Carollia perspicillata*.** Factorial ANOVA analysis was used to examine the effect of dose (PBS and 10 mg/kg LPS), ambient temperature (27˚C and 33˚C), feeding regime (ad libitum and food restricted) and the effects of their interactions on body temperature change.
(PDF)

**S4 Table. Body temperature changes in *Carollia perspicillata*.** Hourly body temperature changes (ΔTb) after immune challenge in *Carollia perspicillata* kept at different ambient temperatures (27˚ and 33˚C) and feeding regimes (ad libitum and food restricted).
(PDF)

**S5 Table. Time of maximum increase of ΔTb in *Carollia perspicillata*.** Time at which the LPS-challenged groups reached the maximum increase in ΔTb.
(PDF)

## Acknowledgments

We thank Augusto G. Paulino, Pedro Henrique Miguel and Ayrton Nascimento for helping us in the field and/or in the performance of experimental tests. All experimental tests were carried out within the facilities of the Laboratório de Fisiologia Animal (LaFA), located at

Departamento de Biodiversidade, Instituto de Biociências, Universidade Estadual Paulista Júlio de Mesquita Filho, Rio Claro, São Paulo, Brazil.

## Author Contributions

**Conceptualization:** Matheus F. Viola, L. Gerardo Herrera M., Ariovaldo P. Cruz-Neto.

**Data curation:** Matheus F. Viola.

**Formal analysis:** Matheus F. Viola, L. Gerardo Herrera M., Ariovaldo P. Cruz-Neto.

**Funding acquisition:** L. Gerardo Herrera M., Ariovaldo P. Cruz-Neto.

**Investigation:** Matheus F. Viola, L. Gerardo Herrera M., Ariovaldo P. Cruz-Neto.

**Methodology:** Matheus F. Viola, L. Gerardo Herrera M., Ariovaldo P. Cruz-Neto.

**Project administration:** L. Gerardo Herrera M., Ariovaldo P. Cruz-Neto.

**Resources:** L. Gerardo Herrera M., Ariovaldo P. Cruz-Neto.

**Supervision:** L. Gerardo Herrera M., Ariovaldo P. Cruz-Neto.

**Validation:** L. Gerardo Herrera M., Ariovaldo P. Cruz-Neto.

**Writing – original draft:** Matheus F. Viola.

**Writing – review & editing:** L. Gerardo Herrera M., Ariovaldo P. Cruz-Neto.

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
