## [Decision Letter · Decision Letter 0]

2 Feb 2024

PONE-D-23-41793COMBINED EFFECTS OF AMBIENT TEMPERATURE AND FOOD AVAILABILITY ON INDUCED INNATE IMMUNE RESPONSE OF A FRUIT-EATING BAT (CAROLLIA PERSPICILLATA)PLOS ONE

Dear Dr. Herrera M.,

Thank you for submitting your manuscript to PLOS ONE. After careful consideration, we feel that it has merit but does not fully meet PLOS ONE’s publication criteria as it currently stands. Therefore, we invite you to submit a revised version of the manuscript that addresses the points raised during the review process.

We look forward to receiving your revised manuscript.

Kind regards,

Jorge Marin Mpodozis, Ph.D.

Academic Editor

PLOS ONE

Journal Requirements:

“This study was supported by a grant to A. P. C. N and L. G. H. M. from Fundação de Amparo à Pesquisa do Estado de São Paulo (FAPESP Visiting Research Program #2017‐17607‐ 6).  A. P. C. N.  was supported by a grant from Fundação de Amparo à Pesquisa do Estado de São Paulo (FAPESP, Grant #2014/16320-7), and L. G. H. M. by a grant from the PASPA‐DGAPA program of the Universidad Nacional Autónoma de México (#814-2018). M. F. V. was supported by a grant from the Coordenação de Aperfeiçoamento de Pessoal de Nível Superior Tecnológico (CAPES, Grant #88882.434214/2019-01).”

Additional Editor Comments:

Your manuscript has been reviewed by an experienced reviewer and by me. We both agree in that the paper is technically sound, and highly interesting, as it sits in the intersection of global environmental changes, comparative immune physiology and zoonosis risk. There are, however, some concerns expressed by the reviewer that need to be addressed before final acceptance.

Reviewers' comments:

Reviewer's Responses to Questions

**Comments to the Author**

1. Is the manuscript technically sound, and do the data support the conclusions?

Reviewer #1: Yes

2. Has the statistical analysis been performed appropriately and rigorously? 

Reviewer #1: Yes

3. Have the authors made all data underlying the findings in their manuscript fully available?

Reviewer #1: Yes

4. Is the manuscript presented in an intelligible fashion and written in standard English?

Reviewer #1: Yes

5. Review Comments to the Author

Reviewer #1: In this study, Viola and collaborators evaluated how the isolated and combined effects of changes in ambient temperature and food availability affect selected physiological and behavioral components associated with the innate immune system of fruit-eating bats (Carollia perspicillata) including fever, leukocytosis, neutrophil/lymphocyte ratio and bacterial killing ability of the plasma. How the immune system deals with the exposure to new pathogens that will occur with global changes, might affect the interaction between wildlife hosts and pathogens and, thus, can impact the dissemination of emerging diseases and zoonoses. In particular, bats are reservoirs of a number of infectious diseases and understanding how global warming and land-use changes might affect their capacity to cope with zoonotic pathogens is important to understanding the dissemination of emerging infectious diseases. The authors observed no changes in food intake, total white blood cell count and bacteria killing activity after challenge with LPS. However, they observed two significant changes; one in the time to reach the highest increase in body temperature under food restriction and second; the magnitude of increased neutrophil/lymphocyte ratio under food restriction and increased ambient temperature.

The experiments performed are well detailed and rigorous and the parameters analyzed reflect the state of the art regarding the analysis of the immune cell components in non-rodent mammals. In this regard, I only have some few comments on this manuscript:

Could the authors detail why they only considered the effect of changes in ambient temperature and food restriction for only two days? Why the authors analyzed all the parameters only 24 hours following LPS injection? Some parameters of the innate immune response and acute phase response will surely change within that window, but others regarding the adaptive immune response might take longer.

The authors indicate that they observed differences in the time to reach the highest increase in body temperature under food restriction and the magnitude of increased neutrophil/lymphocyte ratio under food restriction and increased ambient temperature. Do these two parameters affect the immune performance when bats are challenged with real pathogens? Is it possible to perform cell-culture experiments? Is it possible to measure serum antibodies or the antiviral response in bats?

Minor:

The excessive use of abbreviations in the manuscript makes it very difficult to read

In line 50, the sentence has an extra punctuation

In line 121, please revise the sentence, as it says "In bats, recent reviews suggested that bats respond selectively to LPS..."

6. PLOS authors have the option to publish the peer review history of their article (what does this mean?). If published, this will include your full peer review and any attached files.

Reviewer #1: No

---

## [Author Response · Author response to Decision Letter 0]

5 Mar 2024

Thanks very much for your insightful comments. Below we replied to all points, and we hope that our responses and changes helped to clarify your concerns and improving the manuscript. 

1 - Could the authors detail why they only considered the effect of changes in ambient temperature and food restriction for only two days?

AU: We kept the bats for 48 hours under these conditions because we did not want to compromise bat wellbeing; preliminary work with frugivorous bats showed that longer periods under temperature and feeding extreme conditions might be too demanding. Additionally, we were interested in short-period stress, a condition that might be common in nature. 

2 - Why the authors analyzed all the parameters only 24 hours following LPS injection? Some parameters of the innate immune response and acute phase response will surely change within that window, but others regarding the adaptive immune response might take longer.

AU: This is an interesting idea, and we are aware that longer periods might show different patterns in elements of the immune response, but we were concerned that exposing bats to extreme experiment conditions for longer periods might be detrimental for their health. In our new version, we hope to have made our study goal clearer (line 120 to 139) to justify our approach.

3 - The authors indicate that they observed differences in the time to reach the highest increase in body temperature under food restriction and the magnitude of increased neutrophil/lymphocyte ratio under food restriction and increased ambient temperature. 

A - Do these two parameters affect the immune performance when bats are challenged with real pathogens? 

AU: Again, this is an interesting point worth examining in further studies. To the best of our knowledge, no previous bat study has tested the effect of manipulating ambient temperature and food availability in bat immune response to real pathogens. We hope that future studies explore this idea.

B - Is it possible to perform cell-culture experiments? Is it possible to measure serum antibodies or the antiviral response in bats?

AU: Yes, it is possible. Cell culture, serum antibodies and antiviral response have been done in bats (Dejosez et al. 2023; Banerjee et al. 2020). However, this idea goes beyond our study goals. In our new version, we hope to have made our object of study clearer (line 119 to 138).

Dejosez et al. 2023; https://doi.org/10.1016/j.cell.2023.01.011

Banerjee et al. 2020; https://doi.org/10.3389/fimmu.2020.00026

4 - The excessive use of abbreviations in the manuscript makes it very difficult to read

AU: Thanks for this comment, the excessive use of abbreviations in the text has been corrected, and we kept a few that we deemed would facilitate reading the manuscript.

5 - In line 50, the sentence has an extra punctuation

AU: Thanks, extra punctuation has been removed

6 - In line 121, please revise the sentence, as it says "In bats, recent reviews suggested that bats respond selectively to LPS..."

AU: Thanks, the sentence has been corrected

---

## [Editor Report · Decision Letter 1]

11 Mar 2024

COMBINED EFFECTS OF AMBIENT TEMPERATURE AND FOOD AVAILABILITY ON INDUCED INNATE IMMUNE RESPONSE OF A FRUIT-EATING BAT (CAROLLIA PERSPICILLATA)

PONE-D-23-41793R1

Dear Dr. Herrera M.,

We’re pleased to inform you that your manuscript has been judged scientifically suitable for publication and will be formally accepted for publication once it meets all outstanding technical requirements.

Kind regards,

Jorge Marin Mpodozis, Ph.D.

Academic Editor

PLOS ONE
---

## [Editor Report · Acceptance letter]

5 Apr 2024

PONE-D-23-41793R1 

PLOS ONE

Dear Dr. Herrera M., 

I'm pleased to inform you that your manuscript has been deemed suitable for publication in PLOS ONE. Congratulations! Your manuscript is now being handed over to our production team.

Kind regards, 

on behalf of

Dr. Jorge Marin Mpodozis 

Academic Editor

PLOS ONE